# Results Obtained from a Pivotal Validation Trial of a Microsatellite Analysis (MSA) Assay for Bladder Cancer Detection through a Statistical Approach Using a Four-Stage Pipeline of Modern Machine Learning Techniques

**DOI:** 10.3390/ijms25010472

**Published:** 2023-12-29

**Authors:** Thomas Reynolds, Gregory Riddick, Gregory Meyers, Maxie Gordon, Gabriela Vanessa Flores Monar, David Moon, Chulso Moon

**Affiliations:** 1NEXT Bio-Research Services, LLC, 11601 Ironbridge Road, Suite 101, Chester, VA 23831, USA; treynolds@nextmolecular.com (T.R.); gmeyers@nextmolecular.com (G.M.); 2HJM Cancer Research Foundation Corporation, 10606 Candlewick Road, Lutherville, MD 21093, USA; maxiegordon@bellsouth.net (M.G.);; 3BCD Innovations USA, 10606 Candlewick Road, Lutherville, MD 21093, USA; 4Department of Otolaryngology-Head and Neck Surgery, The Johns Hopkins Medical Institution, Cancer Research Building II, 5M3, 1550 Orleans Street, Baltimore, MD 21205, USA

**Keywords:** bladder cancer, loss of heterozygosity, microsatellite instability, Triplex MSA assay

## Abstract

Several studies have shown that microsatellite changes can be profiled in urine for the detection of bladder cancer. The use of microsatellite analysis (MSA) for bladder cancer detection requires a comprehensive analysis of as many as 15 to 20 markers, based on the amplification and interpretations of many individual MSA markers, and it can be technically challenging. Here, to develop fast, more efficient, standardized, and less costly MSA for the detection of bladder cancer, we developed three multiplex-polymerase-chain-reaction-(PCR)-based MSA assays, all of which were analyzed via a genetic analyzer. First, we selected 16 MSA markers based on 9 selected publications. Based on samples from Johns Hopkins University (the JHU sample, the first set sample), we developed an MSA based on triplet, three-tube-based multiplex PCR (a Triplet MSA assay). The discovery, validation, and translation of biomarkers for the early detection of cancer are the primary focuses of the Early Detection Research Network (EDRN), an initiative of the National Cancer Institute (NCI). A prospective study sponsored by the EDRN was undertaken to determine the efficacy of a novel set of MSA markers for the early detection of bladder cancer. This work and data analysis were performed through a collaboration between academics and industry partners. In the current study, we undertook a re-analysis of the primary data from the Compass study to enhance the predictive power of the dataset in bladder cancer diagnosis. Using a four-stage pipeline of modern machine learning techniques, including outlier removal with a nonlinear model, correcting for majority/minority class imbalance, feature engineering, and the use of a model-derived variable importance measure to select predictors, we were able to increase the utility of the original dataset to predict the occurrence of bladder cancer. The results of this analysis showed an increase in accuracy (85%), sensitivity (82%), and specificity (83%) compared to the original analysis. The re-analysis of the EDRN study results using machine learning statistical analysis proved to achieve an appropriate level of accuracy, sensitivity, and specificity to support the use of the MSA for bladder cancer detection and monitoring. This assay can be a significant addition to the tools urologists use to both detect primary bladder cancers and monitor recurrent bladder cancer.

## 1. Introduction

Microsatellite instability (MSI) is a molecular genotype resulting from genomic hypermutability, and it is initially discovered as genome-wide variations in the length of microsatellite sequences. As part of familiar colon cancer syndrome, MSI was commonly observed among individuals with Lynch syndrome [1,2,3]. Since its initial discovery, MSI has been recognized as a generalized phenomenon among a wide spectrum of sporadic cancers [4,5,6,7,8], and the underlying mechanisms for these cases of sporadic cancers seem to be unrelated to inherited genetic mutations. At the same time, they are based on an epigenetic mechanism, namely the methylation of MLH1. For both inherited and sporadic solid tumors, the most deleterious outcome of MSI would be the accumulation of frameshift mutations in tumor-associated genes, which then can pave crucial pathways leading to human carcinogenesis—the development of cancer. The molecular diagnosis of MSI is relatively simple and can currently be accomplished by examining PCR products amplified with a few (typically five to seven) informative microsatellite markers based on a polymerase chain reaction (MSI–PCR) [9,10].

Bladder cancer, the majority of which is transitional carcinoma of the bladder, is one of the major causes of morbidity and mortalities among elderly patients in the Western world [11]. While our understanding of its etiology, molecular characteristics associated with disease progression, and management guidelines have progressed, early detection is still challenging. It renders bladder cancer one of the critical and ideal candidates for early disease screening [12,13,14,15]. The urine cytology test, one of the most commonly used screening methods, is of limited value due to its low sensitivity, particularly to low-grade tumors, while it does provide high specificity. Cystoscopy has been the gold standard for detecting bladder cancer for decades. However, it is an invasive test and often carries unwanted complications, and its relatively high cost can sometimes limit its use. Although urine cytology and cystoscopy are considered standards of care, they are less than optimal in detecting all forms of bladder cancer. Urinary cytology has a sensitivity of 25–50% and a specificity of 90–100%. Cystoscopy has a sensitivity of 90–100% and a specificity of 75%. Consequently, there is a need to improve the current practice of bladder cancer surveillance. New molecular tests designed to detect both qualitative and quantitative changes among cellular and subcellular alterations exclusively associated with bladder cancer have been explored [14,15,16], and several new “molecular assays” for diagnosing urothelial cancer have been developed over the last two decades.

Overall, the application of several molecular assays has been used in conjunction with traditional screening methods, demonstrating promising results [16,17,18,19,20]. Based on histopathology and clinical presentations, two different types of noninvasive bladder cancer have been classified, and they include the frequently recurring papillary tumor (Ta) and the more aggressive carcinoma in situ (CIS). While either type can progress into invasive tumors (T1–T4), the chance that low-grade Ta tumors progress to invasive disease is much less likely than that of high-grade Ta tumors and CIS. Notably, nowadays, it is commonly accepted that noninvasive tumors have been grouped as “superficial bladder cancer” and that noninvasive (that is, Ta and CIS) tumors are distinguished from invasive tumors, which invade the basement membrane [17,18,19].

A loss of heterozygosity (LOH) is typically identified by comparing the DNA isolated from tumors to germline DNA, such as that isolated from blood, the germline control. This LOH can be detected using a method known as microsatellite instability analysis (MSA) [1,2,3]. LOH involving p16 (chromosome 9p21) and p53 (17p13) is associated with non-muscle-invasive transitional cell carcinoma (TCC), which have, so far known to be the two most common regions of LOH in bladder cancer [20,21,22,23,24,25]. LOH at 9p has been shown to have prognostic value in non-muscle-invasive bladder cancer, while other LOH loci implicated in bladder cancer progression are characterized among different genetic loci, including 18q, 4p, 16, 20, and 21 [26,27,28,29,30,31,32,33,34]. It is important to recognize that microsatellite instability and LOH are two different genomic instability types; one is microsatellite, and the other is chromosomal, and their etiologies are different. For example, in colorectal tumors, tumors with microsatellite instability have completely different paths and outcomes compared to chromosomal instability tumors. Overall, LOH can be detected via a gene scan, which may differ from microsatellite instability analysis [1,2,3]. Abnormalities involving p16 (chromosome 9p21) and p53 (17p13) are associated with superficial transitional cell carcinoma (TCC), and so far, these loci have been the two most common regions of LOH in bladder cancer [20,21,22,23,24,25,26,27]. LOH at 9p has been shown to have prognostic value in non-muscle-invasive bladder cancer. Other loci implicated in the progression of bladder cancer have also been characterized, and among different genetic loci, these involve chromosomes 18q, 4p, 16, 20, and 21 [28,29,30,31,32,33,34,35]. LOH is typically identified by comparing the DNA isolated from tumors to germline DNA, such as that isolated from blood. Short tandem repeat (STR) regions (also known as microsatellite regions) within chromosomes are unstable in cancerous cells and are deleted via errors made during DNA replication, causing LOH in the tumor sample and a loss of heterozygosity (LOH) in the tumor sample. Microsatellite instability analysis (MSA) targets such tandem repeats in genomic DNA and is one of the methods used to evaluate the loss of heterozygosity (LOH) that occurs with tumor cell transformation [24,25,26]. These biomarkers, originally developed at Johns Hopkins University, represent a panel of 15 STRs and have been shown to detect deletions in the DNA isolated from the urine sediment of bladder cancer patients before cystoscopic evidence of a tumor [34,35,36]. Several studies have shown that these microsatellite markers can be profiled in urine to detect bladder cancer cells [37,38,39,40,41,42,43,44]. Various combinations of MSI markers have, so far, resulted in overly sensitive defection for low- and high-grade lesions with sensitivities of 67%, 86%, and 93% for recurrent G1, G2, and G3 lesions, respectively, while such MSA still carries an average specificity of 88% [36,37,38,39,40,41,42,43,44]. Moreover, microsatellite analysis (MSA) could predict bladder tumor recurrence before cystoscopically detected evidence in all studies with extended follow-ups [44]. Although MSA has extremely high potential, several important issues and concerns have been raised and need to be resolved before its use in everyday clinical practice. These, including establishing assay automation, completing multi-center studies, and establishing guidelines excluding patients with persistent leukocyturia, must be addressed appropriately if this analysis is to become clinically important.

The discovery, validation, and translation of biomarkers for the early detection of cancer is the primary focus of the Early Detection Research Network (EDRN), an initiative of the National Cancer Institute (NCI). A prospective study sponsored by the EDRN was undertaken to determine the efficacy of a novel set of MSA markers for the early detection of bladder cancer [45,46]. At the time of the selection of the Johns Hopkins MSA panel for the EDRN study, which occurred between 2003 and 2008, the 15 markers selected were the best studied markers and had been reported in the literature in smaller studies. More importantly, these markers were accepted by the EDRN committee as the panel for testing after the process of a literature review. This work and data analysis were performed by collaborating with academics and industry partners. Out of 500 individual participants in the study (300 cancer patients and 200 benign controls), a summary analysis of samples, exclusive of various confounding variables and sample errors, showed ~80% sensitivity and specificity in cancer patients compared to healthy controls [45].

Microsatellite analysis for the detection of cancer faces several significant technical challenges, specifically in allele calling and interpretation. Lab-to-lab variability due to instrumentation and personnel has created differences in assay performance. Additionally, both stochastic effects and variations in peak height effects between the DNA derived from urine sediment samples and the DNA derived from blood samples have provided more variation in the results, which added to interpretation differences. These differences have been especially evident in samples with which MSA results produced results slightly above or below the cut-off ratios established for LOH. Hence, the qualification of STR assays presents significant challenges, and establishing the right parameters to be used for determining LOH from potential tumor cells isolated from urine sediment has become one of the key hurdles in establishing a dependable MSA test [34,35,36,37,38,39,40,41,42,43,44].

Recently, as a way to develop standardized and less costly MSA for the detection of bladder cancer, we developed a triple-multiplex-PCR-based MSA assay. In this report, we present an overview of the assays used to analyze EDRN samples. We also present an overview of the validation trial design. Importantly, we undertook a re-analysis of the primary data from the Compass study in order to enhance the predictive power of the dataset in bladder cancer diagnosis. Using a four-stage pipeline of modern machine learning techniques, including outlier removal with a nonlinear model, correcting for majority/minority class imbalance, feature engineering, and the use of a model-derived variable importance measure to select predictors, we were able to increase the utility of the original dataset to predict the occurrence of bladder cancer (Table 1).

The area under the ROC Curve (AUC) is a metric that can be used with any predictive model. An AUC range of 0–1 indicates the following:

Empirically measured via cross-validation;Derived from the false positive and false negative rates;An AUC of 0.7 is the threshold for an “acceptable” model;An AUC of 0.8 is the threshold for a “good” model;An AUC of 0.9 is the threshold for an “excellent” model.

## 2. Results

### 2.1. Development of Different Models

To model the relationship between MSA markers and cancer/normal status, we used the machine learning technique of stochastic gradient boosting (SGB). This modeling algorithm is a member of the larger class of ensemble models, in which many simple models (classification trees) are combined as an aggregate to approximate a more complex and powerful model. SGB uses an iterative approach to fitting a dataset in which each step adds a new model to the ensemble created to reduce the classification error (residual) of the previously existing ensemble of models. In this way, more of the resources (simple classification trees) are allocated to the more difficult-to-classify cases in a dataset, and the overall performance of the ensemble is improved. This relationship is expressed as the weighted average of weak learners in which alpha_t is the weight calculated by considering the last iteration’s error (Equation (1)).

To evaluate the performance of SGB in predicting cancer/normal samples from the MSA dataset, 80% of the cases were randomly assigned to a training set, and 20% of the cases were randomly assigned to a held-out test set. K-fold cross-validation was used to tune six parameters of the SGB model in the training set, which was then used to predict cancer/normal status in the test set. The performance was evaluated using classification accuracy and the ROC area under the curve (AUC). AUC is a measure of the relationship between false positives and false negatives, and it provides a more robust estimate of model performance even when class membership is unbalanced in a dataset. AUC varies from 0 to 1, and the level of 0.8 is often used as the threshold for “good” performance.
(1)$$f\left(x\right)=\overset{t}{\underset{i=1}{\sum }}a_{t}h_{t}\left(x\right)$$

#### 2.1.1. Model 1: Feature Engineering

Feature engineering in machine learning is defined as a method to transform predictive variables from a dataset using domain knowledge to create a more realistic model. Feature engineering was used as the process of transforming raw data into features that were suitable for machine learning models. In other words, it is the process of selecting, extracting, and transforming the most relevant features from the available data to build more accurate and efficient machine learning models. Bladder cancer can be predicted not only through the presence of specific MSA markers but also through the overall burden of positive MSA markers present. To represent the total MSA marker burden in a sample, we encoded the total number of positive markers present for each sample as a separate predictive variable and evaluated the impact of this derived feature on model performance. Adding the total count of positive markers for each case increased the AUC from 0.674 in the base model to 0.702 in the model with its marker count encoded as a separate variable. The accuracy was unchanged (Figure 1).

#### 2.1.2. Model 2: Outlier Removal Using a Non-Linear Model

To remove outliers from the model, we used random forest, an ensemble machine learning algorithm based on classification and regression trees. The proximity measure of random forest was previously used to identify outliers in a regression model by Riddick. The proximity matrix between cases in random forest is calculated by determining the frequency at which two different cases are assigned to the same terminal node of a classification or regression tree (Equation (2)). With this approach, cases lying outside the boundary by more than one standard deviation defined by the proximity matrix between cases were removed from the model, and this procedure increased the performance and robustness of the model.
(2)$$f\left(x\right)=\overset{t}{\underset{i=1}{\sum }}a_{t}h_{t}\left(x\right)$$

Using the proximity matrix of the random forest algorithm to remove outliers from the MSA dataset resulted in 193 cases of combined cancer/normal samples or a reduction of 5.3% in number from the original dataset used in the current study. Removing the outliers increased the AUC to 0.80 and the accuracy to 0.872 (Figure 2).

#### 2.1.3. Model 3: Addressing Class Imbalance

Class imbalance occurs in machine learning problems in which classes of observation labels are unequally represented. Typically, class imbalance reduces model performance because a disproportionate amount of model resources is allocated to the majority class, leading to a much lower accuracy/recall for the minority class. Because the MSA dataset has a 3:1 class imbalance between the cancer samples and normal samples, we addressed this class imbalance using the method of minority class up-sampling. Through this technique, the samples are first randomly assigned to a training set and a test set. The samples from the minority class in the training set are up-sampled by drawing samples randomly from the minority class, with replacement, until the frequency between the minority class samples and the majority class samples in the training set is equalized. Normalizing class frequency in the MSA dataset increased the AUC to 0.812, while the accuracy slightly decreased (Figure 3).

### 2.2. Using Variable Importance to Select Predictors

Although univariate statistical tests can be used to rank the importance of variables in a dataset, more advanced ranking metrics using model-based importance may provide more useful information about the predictive importance of features used in a machine learning model. Methods such as random forest, gradient-boosted regression trees, and multivariate adaptive regression splines generate variable importance as a natural consequence of how their algorithms function. Permutation-based variable importance, in contrast, can be combined with any type of computational model, and it measures the increase in the model prediction error that results from the individual permutation of values among the input features. This relationship is formally represented by Equation (3), in which the importance, ij, for feature fj is defined for all K repetitions multiplied by the summation of each repetition where Sk,j is the score computed on the permuted data, D*kj*.
(3)$$i_{j=}s-\frac{1}{K}\overset{K}{\underset{k=1}{\sum }}S_{k.j}$$

Selecting the top-ranked eight MSA markers generated via permutation-based variable importance increased the AUC to 0.829 and the accuracy to 0.891. (Figure 4, Table 2).

### 2.3. Model Stacking

Unsupervised learning using principal component analysis (PCA) and principal component surfaces (PCS) was applied to analyze both the cancer and normal samples in the MSA dataset (Figure 5). The results showed that the class boundaries between the groups were not clearly defined. To improve the performance in class discrimination using supervised learning, we tested the performance of single-class prediction using a support vector machine with a radial kernel. Single-class prediction models only examine the core identity of a single class at a time and identify anomalies as combinations of input feature values on the extreme edge of this feature space. Using this approach, single-class predictions from both the cancer and normal samples were combined via model stacking, which combines predictions from disparate model types in the first layer with a higher-level model in the second layer. In this case, elastic net logistic regression was used to combine first-level predictions from both the single-class models and the previously created SGB model to create a new ensemble model that improved performance (Figure 5 and Figure 6).

## 3. Discussion

While many aspects of clinical presentations, including the expected clinical course of disease progression, make bladder cancer a potential screening target, the common consensus at this point, based on several careful analyses, is screening high-risk populations [13,14,15,16,17], not general populations. Indeed, there are two key considerations for which bladder cancer screening will be important in the upcoming decades. Foremost, the persistently high prevalence of smoking in the last several decades is expected to serve as a key hazard for long-term urothelial carcinogenic effects for the upcoming several generations. Moreover, as bladder cancer usually cannot metastasize before it becomes locally invasive [19,20,21], this presents a valuable opportunity for the early detection of bladder cancer in a window of time between a tumor’s origination and invasion. While, on average, 70–80% of bladder cancers are noninvasive, groups with noninvasive or organ-confined invasive tumors (T1/2, N−) have higher survival rates than those with more advanced-stage tumors, including extravesical tumors (T3/4, N−), lymph node metastases (any T, N+), or metastases to distant organs (any T/N, M+) [12,13,14,47,48,49,50,51,52,53,54]. It is unfortunate to recognize that, while there have been numerous efforts for constant improvements in the management of these more advanced diseases using chemotherapy or chemoradiation therapy, the overall management outcome of bladder cancer has not been significantly improved over the past three decades [1,2,3]. In summary, patients with organ-confined disease will face a 92.5% survival rate for five years, while the current five-year survival rates for patients with extravesical disease/nodal metastasis and distant metastasis are much lower, at 44.7% and 6.1%, respectively. These findings suggest that, once bladder cancer progresses beyond a certain stage, like organ-confined disease, successful management becomes significantly more difficult. Therefore, the early detection of bladder cancer has been proposed to be the most effective way to improve its management outcomes. Indeed, the management of noninvasive cancers does not usually depend on aggressive treatment measures like cystectomy, systemic chemotherapy, or chemoradiation therapy, and therefore, it is associated with fewer treatment-related morbidities and is more effective than the aggressive treatment for invasive tumors [16,47,48,49].

MSA for cancer detection faces several significant technical challenges, specifically in allele calling and interpretation. This study’s results mirror those obtained by Butler et al. during validation studies for STR analysis in their research on human identity. Most importantly, variability from lab to lab due to instrumentation, personnel skills, and operation protocols led to differences in assay performance. Moreover, both stochastic effects and variations in peak height effects between the DNA derived from urine sediment samples and DNA derived from blood samples resulted in more variation in the final reports, adding to the interpretation differences. These differences were especially problematic for testing samples that produced results slightly above or below the cut-off ratios established for LOH. Hence, establishing the parameters for determining LOH from potential tumor cells isolated from urine sediment becomes an essential challenge in properly qualifying an STR assay. LOH is ordinary in human solid tumors, and it allows the expressivity of recessive loss-of-function mutations in tumor suppressor genes [22,23]; therefore, detecting recurrent LOH in a genomic region can provide critical clues and future guidance for the localization of tumor suppressor genes. Multiple factors need to be considered for a proper interpretation of LOH. For example, most clinical samples collected from urine will contain a mixture of tumor and normal cells, producing a potential mixture at each locus being analyzed, and this can then obscure losses of genetic material resulting in tumor cells, while LOH can only be determined via the comparative analysis of a control profile obtained from a blood sample. Recently, our group successfully reproduced 15 dependable markers based on three multiplex PCR assays. Overall, this report illustrates the challenge of qualifying a technically complex biomarker assay that requires interpretation by an analyst.

So far, many independent groups have confirmed that MSA achieves superior sensitivity (75–96%) compared to cytology (13–50%) in various clinical settings [23,24,25,26,27,28,29,30,31,32,33,34,38,39,40,41,42,43,44,45,46,47,48,49,50,55,56,57,58,59,60]. In summary, our careful analysis based on 377 cancer samples from 1997 to 2001 reports suggests that the sensitivity of MSA was 90 percent and that specificity was 100 percent [31,32,33,34,35,36,37,38,39,40,41]. Importantly, unlike conventional cytology [50,51,52,53,54], it appears that microsatellite analysis (MSA) can detect low-grade and low-stage disease as accurately as high-grade and high-stage disease [36,37,38,39,40,41,42,43,44,55,56,57,58,59,61,62,63,64,65,66,67,68]. Frigerio et al. [37] published one of the first exciting reports suggesting that the combined use of cytology and LOH analysis resulted in higher sensitivity than either test alone in identifying primary tumors and that it could detect almost all recurrent diseases in voided urine. Moreover, van Rhijn et al. [43] demonstrated that, for 47 Caucasian patients with confirmed non-muscle-invasive bladder TCC (37 pTa and 10 pT1) at their initial diagnoses, MSA could potentially provide an efficient early screening tool. In summary, this report provided three crucial clinical observations. First, MSA correctly identified 94% of primary tumors (44/47). Second, MSA correctly identified 92% of tumor recurrences (12/13). Third, MSA predicted the chance of future recurrences, as 75% of tumor recurrences (9/12) were molecularly detected 1–9 months before cystoscopic evidence of recurrent disease. Likewise, several important studies have demonstrated that urine microsatellite analysis reliably detects non-muscle-invasive bladder tumors and can be used as a reliable test for detecting and predicting tumor recurrences before cystoscopic evidence of recurrent disease, and these studies have also shown that such detection rates could be improved when this analysis was offered in combination with urine cytology. In this sense, it is exciting to recognize that the meta-analysis by van Rhijn et al. reported one of the first extensive literature reviews for the use of 18 markers and concluded that MSA, ImmunoCyt, NMP22, CYFRA21-1, LewisX, and FISH are the most promising markers for the surveillance [67,69,70,71] and early detection of bladder cancer. Nevertheless, two points must be carefully considered for this biomarker-based bladder cancer detection. First, insufficient clinical evidence warrants the substitution of the cystoscopy follow-up scheme for any currently available urine marker tests. Second, the data so far are inconsistent with the sole use of molecular tests in patients facing a high risk of developing bladder cancer. In summary, many studies have shown that various molecular tests offer the best value in improving the diagnostic accuracy for high-risk population groups in the initial diagnosis of bladder cancer and predicting recurrence only when used in conjunction with cytology and cystoscopy. It was further suggested that molecular testing can reduce the need for these procedures with cytology and cystoscopy. Again, many studies have shown molecular tests to have value in not only improving the diagnostic accuracy of high-risk population groups in the initial diagnosis of bladder cancer but also predicting recurrence, while such improved diagnostic accuracy could be persistently observed only when used in conjunction with cytology and cystoscopy. From a genome-wide perspective, MSI variation is not prevalent in bladder cancer (BC), especially early BC. This is a well-known limit to this kind of biomarker in BC detection. Therefore, future efforts need to focus on MSA markers in the early stage of BC development.

Several points warrant further discussions to enhance both the sensitivity and specificity of MSA [69,70]. First, establishing a robust genetic marker profile and determining, for each of the analyzed microsatellites, individual threshold values for an LOH/allelic imbalance can maximize the sensitivity of tumor DNA detection without compromising its specificity. In this sense, we used rigorous statistical analysis based on reported high-quality published studies, as shown in Table 2. Second, a careful setup of dependable methods is necessary to avoid erroneous LOH judgments due to PCR artifacts, which have also been repeatedly described by others [41,42,43,44,69,70]. Third, performing MSA on a genetic analyzer, which is commonly used nowadays, must always be a standard approach to MSA practice, as it can incorporate two major advantages. First, sample processing can be largely automatized, and the results are provided in the form of a numerical data readout, independent of the inter-observer variability associated with the complex interpretation of morphological features. Moreover, the determination of LOH ratios on this platform is highly reproducible, and it provides reliable results even in situations when the cell conservation is suboptimal for cytological evaluation and/or FISH. In fact, in our experiences, the use of a standard genetic analyzer like an ABI 3100 machine provided MSA with a significant advantage, as MSA was little affected by changes in PCR conditions or amounts of genomic DNA applied, as long as it met the minimum requirements. For example, when we repeated multiple rounds of our triplex PCR, most of the markers resulted in consistent findings. Importantly, the amount of non-degraded measurable genomic DNA to be analyzed needs to be sufficient, and we propose using at least 20 to 30 ng of urine genomic DNA to be on the safe side of DNA quantity, 20 ng for 10 markers, and 30 ng for 15 markers. We used DNA amounts based on this guideline. There were a significant number of patient samples present in the pivotal study that did not meet this threshold and had to be removed from our analysis. Urine samples in particular are prone to DNA degradation, and sample collection and post-collection handling and storage are key to the preservation of sample quality. In fact, when the DNA extracted from urine samples was less than 30 ng, the result of MSA was either inconsistent or non-reproducible. Notably, while 100 patients participated in Group 1, due to the insufficient amount of measurable DNA samples, only 94 out of 100 samples were suitable for the final analysis. Likewise, in the Group 3 cohort, due to the insufficient amount of measurable DNA, 236 out of 300 samples were suitable for the final analysis. This prospective study was designed to determine the efficacy of the panel of MSA markers for the detection of bladder cancer. Notably, the Group 2 cohort experienced various but significant benign conditions (the confounding group) for which the panel has been well known for its inaccuracy in detecting primary or recurrent bladder cancer through the gold standard (cystoscopy or cytology). These conditions include gross hematuria, indwelling catheters, stones, and apparent urinary tract infections. Most of the time, cystoscopies are known to be less accurate, usually in the 65% to 75% range, for these confounding conditions. Because of this, for these acute benign conditions, as a standard practice, cystoscopy is usually performed after the resolution of the confounding conditions. In our trial, Group 2 was studied to see whether the MSA test at least could achieve similar accuracy with these confounding conditions, and MSA did achieve concordant or even better results (78% accuracy). This may be due to the presence of blood in the urine samples, which provides less DNA in the urine component to be examined. Hence, the Group 2 samples probably mirrored the MSA results from the blood sample comparator, producing a negative or normal result. Therefore, patients with confounding conditions are not recommended for MSA analysis, and they most likely favorably skewed the accuracy identified during this analysis.

Finally, statistical analysis using machine learning was applied to four different models, providing insight into the accuracy of the MSA method used in this pivotal study. It should be stressed that this analysis was beyond the scope of the original concept for the analysis of this pivotal study and was only performed blinded in Model 1, and then it was applied unblinded to the subsequent models (Models 2–4), resulting in the manipulation of the data set to be analyzed to provide the most appropriate version for statistical analysis. The most appropriate data for analysis included the sets in Model 2, the removal of outlier samples, and Model 4 (the eight top markers only), which included gradient boosting. These results were summarized in Table 1, and they show the base model’s accuracy in predicting bladder cancer in the sample to be 83%, with the accuracy improving in the subsequent models, Model 2 (87%) and Model 4 (90%). Model 3, which included class rebalancing, did not improve accuracy significantly (83% to 85%) alone. Model 4, which only analyzed the top eight informative markers (Table 2) from each sample, was also beyond the scope of the original concept for the analysis of the pivotal study, but it proved to achieve the highest level of accuracy (90%). This may have been due to the removal of data from the least informative MSA markers, which resulted in an increase in the accuracy of the assay by lowering the level of false positive or false negative results that would have been present if those markers had been included in the data set.

The original goals of the EDRN study were accuracy, sensitivity, and specificity rates above 90%. Hence, the study did not meet those requirements to achieve sufficient validation. This was primarily due to several flaws in the study design, including the lack of accounting for the overall homozygosity of each marker found in the patient population (which was high, at 20% per patient), sample degradation issues due to longer cold storage than anticipated, and a lower sample DNA concentration, especially from normal urine samples. Nonetheless, the statistical analysis presented in this study took into account these design flaws and, once they were addressed, the MSA assay nearly achieved the stated EDRN goal for validation with a range of 80–90% accuracy, depending on the statistical model applied. This is significant because this assay achieved the highest accuracy rate when compared to other known methods, including cystoscopy and urovision analysis. Hence, if MSA is used as an additional tool for a urologist, the MSA test can help more accurately detect bladder cancer. Moreover, if used in recurrent bladder cancer monitoring, this assay can significantly lower the cost of surveillance due to its lower cost compared to cystoscopy. In addition, since it is a less invasive test approach using urine and blood as a sample, MSA can not only increase patient compliance for monitoring but also cut down on unnecessary procedures.

In conclusion, we have presented data for the development of our MSA assay covering 15 markers, and we further propose that our assay is a potentially time- and cost-effective genetic assay for bladder cancer detection with accuracy as high as 85%. In this paper, we have also discussed various prior data regarding MSA-based bladder cancer detection. These results are consistent with the multiple other reports analyzing smaller samples that we have discussed above in terms of diagnostic accuracy. These results strongly support the use of MSA as a tool for the detection of bladder cancer to improve the management of bladder cancer.

## 4. Materials and Methods

### 4.1. Matched Blood and Urine Genomic DNA Samples

Blood and urine specimens for the initial qualification study were obtained from Dr. David Sidransky’s lab at Johns Hopkins University (the JHU sample). These included 15 biopsy-proven superficial bladder cancer cases, 10 healthy controls, and participants in the study on assay development for MSA that we have reported here, all of whom were deemed eligible for the study. These samples were previously used for the determination of a proper cut-off value for a series of 10 MSA markers. The parent trial and the secondary analysis reported in this manuscript were approved by the Institutional Review Board from Johns Hopkins University. Written informed consent was obtained from all participants and/or their legal guardians. All research was performed in accordance with relevant guidelines/regulations from Johns Hopkins University. The research involving human research participants was performed in accordance with the Declaration of Helsinki.

### 4.2. EDRN Study

The prospective study was designed to determine the efficacy of the panel of MSA markers for the detection of bladder cancer and bladder cancer recurrence using the developed clinical MSA assay [45,46].

For disease-free study populations, Group 1 included 100 cases with no histories and with normal urine and cytology. Group 2 included 100 cases with 1 of 4 urologic processes* requiring cystoscopy, which, in the past have achieved confounding results with urinary tumor detection assays. For the bladder cancer study population, Group 3 included 300 patients with bladder cancer, both incident and recurrent. The study schema is shown below (Figure 1):

EDRN Study 1.0 Objectives

Goal 1: To determine the sensitivity and specificity of the microsatellite analysis (MSA) of urine sediment using a panel of 15 microsatellite markers when detecting bladder cancer in participants requiring cystoscopy. This technique was compared to the diagnostic standard of cystoscopy, as well as to urine cytology.

Goal 2: To determine the temporal performance characteristics of the microsatellite analysis of urine sediment.

Goal 3: To determine which of the 15 individual markers or which combination of markers that make up the MSA test are most predictive of the presence of bladder cancer.

As part of the study, a quality control CLIA/College of American Pathology (CAP) accredited laboratory, (QA Lab) University of Maryland Baltimore Biomarker Reference Laboratory (UMB-BRL), performed quality control analysis on 10% of the samples assessed by the testing laboratory (the QC samples). The parent trial and the secondary analysis reported in this manuscript were approved by the Institutional Review Board of EDRN. Written informed consent was obtained from all participants and/or their legal guardians. All research was performed in accordance with the relevant guidelines/regulations from EDRN. The research involving human research participants was performed in accordance with the Declaration of Helsinki. The identification of the sample’s status was indicated by EDRN as cancer or normal after the results were evaluated.

### 4.3. DNA Extraction and Quantification

Matched blood and urine samples remained de-identified during the study. Genomic DNA from the buffy coat of blood collected in ACD collection tubes was purified at the testing lab using the QIAGEN QIAMP Blood Mini kit according to the manufacturer’s instructions. Genomic DNA from urine sediment was purified using the QIAGEN QIAMP Viral RNA Mini kit. The DNA was quantified against a standard curve of human DNA using TaqMan β-actin Detection Reagents from Applied Biosystems. In this study, 30 mL of urine samples and 5 mL of white blood were used.

### 4.4. STR Targets and PCR Primers

The panel of STR targets for the MSA assay was originally developed at Johns Hopkins University as a radioactive PCR assay [34]. The testing lab converted the radioactive PCR assay to a high-throughput capillary electrophoresis assay of fluorescent PCR products. The resulting assay comprised two or three multiplex reactions, of which primers were selected from the report in Table 2, and each primer sequence of PCR product is outlined in Table 3. The 5′ fluorescent primers were obtained from Applied Biosystems. The 3′ primers were obtained from Integrated DNA technologies. Following Round 3 of the qualification studies, the single plex reaction was removed from the analysis due to poor performance in the assay. The primers were initially mixed at an equimolar concentration (2 pm for each primer); following refinement of the assay, the primer concentrations were further adjusted. The design of five sets of multiplex PCR was previously initiated from the assay developed for an EDRN study [45,46].

### 4.5. Selection of 15 Markers

Out of the 9 selected publications shown in Table 3, we selected 15 different MSA primer sets for 16 markers, as shown in Table 4. These primer sets had repeatedly shown dependable high LOH in bladder cancer samples with an exceptionally low number of LOH among samples from healthy individuals. Among a total of over 350 patients, the combined data achieved over 90% sensitivity and 100% specificity. These 16 different markers were characterized for separate independent PCR-based MSA analysis for the analysis of desirable cut-off values.

### 4.6. Development of Triplet Multiplex PCR

We originally designed triplet, three-tube, multiplex-PCR-reaction and singlet, one-tube, single-plex-PCR primer sets, and the compositions of the individual primers used for triplet (three-tube) multiplex PCR are described in Table 5. This primer, D13S304 (AAAG)n 6-FAM GGCTGCATGAGCCCTAAGTA TGGGTGACACAGTGAGACTCTA), was not included in the final triplet MSA assay since this marker had resulted in inconsistent PCR amplification. Notably, the MP2 marker MBP generated two distinct amplifications that were included in the results as marker MBP and MBPA, making a total of 15 amplified markers from 14 primer sets in the multiplex. The cut-off values were previously designed and described in Table 6 and Table 7. This final triplet assay we then repeated multiple times with various samples, including the original 25-cancer-case sample and the 15-healthy-control sample (the JHU samples), and the reproducibility was consistent. Finally, we performed, in a non-blind fashion, a test of the 20 QC samples from the EDRN study using our newly established assay. Of these QC samples, 6/20 cancer samples were determined to be cancer, based on an overall evaluation of all 15 markers tested, and 14/20 samples from the healthy individuals were determined to be non-cancerous.

### 4.7. EDRN Pivotal Validation Study

The prospective study was designed to determine the efficacy of the panel of MSA markers for the detection of bladder cancer and bladder cancer recurrence using the developed clinical MSA assay [45,46], and more than 10 different institutes from the US and Canada participated in this study. Three groups of different patient populations were sampled for this study, as outlined in the Materials and Methods section.

### 4.8. Statistical Analysis of the MSA Clinical Trial Results

The MSA dataset was produced using a multi-institute study of the detection of bladder cancer via microsatellite analysis. This study contained diagnostic data for 15 MSA markers determined via multiplex PCR analysis from both blood and urine samples. The primary cancer group comprised 300 patients with incident or recurrent superficial bladder tumors, while the control group consisted of 2 groups of 100 patients (Table 8). The first control group of 100 patients without a history of genitourinary diseases was used in a re-analysis of the dataset, while the second group was not used.

#### 4.8.1. Data Cleaning and Normalization

The original matrix of 15 MSA marker results for 300 cancer patients and 100 controls, called Model 1 or the base set, was reduced by eliminating the degraded or unanalyzable samples that contained data with the designation “Insufficient Peak Height” for those that had five markers or had five sites or homozygous sites or a combination of both. These were called non-informative markers, and samples with 5 or more were detrimental to the analysis. The resulting data for 317 samples were treated as a four-level factor with the designations of “Negative”, ” Non-Evaluable”, “Non-Informative”, and “Positive”. Then, 243 cancer samples and 74 normal samples were left from the original set. For further analysis, we encoded the 15 MSA markers for each sample in a binary fashion, with “1” indicating a positive result and “0” indicating a non-positive result.

#### 4.8.2. Fitting a Model Using Stochastic Gradient Boosting

The xgboost R package and the tidymodels framework were used to fit a model, Model 2, to the data. A vector of 100 true random numbers was first generated using random.org. This list of numbers was used as a collection of random seeds to partition the data into 100 sets of training samples (80%) and test samples (20%). Within each training sample, six parameters (tree_depth, min_n, loss_reduction, sample_size, mtry, and learn_rate) were tuned using leave-one-out cross-validation. The resulting parameterized model was then fit to the test set. Both the accuracy and the area under the ROC curve (AUC) were evaluated.

#### 4.8.3. Feature Extraction Using a Variable Importance Metric

The R package vip was used to rank MSA markers by importance in the model referred to as Model 3 in this paper. Variable importance was measured as the increase in RMSE from the fitted model that resulted from permuting an input variable value with another randomly selected variable value in the input matrix. This process was repeated 10,000 times while measuring the mean increase in the RMSE for each of the MSA markers. The importance values were then used to rank the MSA markers. To test whether the use of the high-importance variables could increase the performance of the model, versions of the dataset were created that contained the top 4, top 6, and top 8 MSA markers, and the model results were compared to those of the dataset with all 15 markers.

#### 4.8.4. Selecting the Optimal Number of MSA Markers

Model 4: The eight MSA markers that were ranked as the most important were selected because these markers achieved the same classification accuracy as the 15 MSA markers (0.766, Figure 5 and Figure 6) with the highest AUC (0.743 vs. 0.718).

### 4.9. Multiplex PCR: Triplet (Three-Tube) Multiplex PCR

We ran five sets of optimization reactions (qualification) to design a triplet, 3-tube PCR, multiplex PCR reaction using the primers shown in Table 5. For the first three rounds of qualification, the STR regions were amplified using AmpliTaq Gold DNA Polymerase (Applied Biosystems). The PCR conditions were 4 mM of MgCl2/0.2 mM of dCTP/0.2 mM of dGTP/0.4 mM of dUTP and 0.5 mM of dATP/2 units Amplitaq Gold. MP1 and MP2 were amplified using 4 ng of total genomic DNA, respectively, for both the blood and urine DNA, and MP3 and the D13S804 Singleplex reaction were amplified using 6 ng of blood or urine DNA in a 25 µL final volume. The PCR cycling conditions were 95 °C for 11 min/32 cycles of 94 °C 1 min, 55 °C for 1 min, and 72 °C for 1 min/60 °C for 45 min/4 °C hold. For Rounds 4 and 5 of the qualification, the PCR amplification conditions were modified to utilize a 1X FastStart TaqMan Probe Master (Roche, Basel, Switzerland). The DNA concentrations and the PCR cycling conditions remained the same.

### 4.10. Capillary Electrophoresis

Capillary electrophoresis was performed on 1 µL of PCR product using the 3100 genetic analyzer from Applied Biosystems (Foster City, CA, USA). Matched blood and urine samples were run in the same injection to prevent differences in run conditions between the matched specimens. Following electrophoresis, the data were analyzed using Gene Mapper v3.1 (Applied Biosystems), and the peak sizes and peak heights were exported as a tab-delimited text file.

### 4.11. Calculation of the Loss of Heterozygosity

The data exported from the 3100 analyzer were imported into Microsoft Excel for analysis. In order for the data to be acceptable, the positive control DNA (HL-60 genomic DNA (ATCC)) had to meet strict peak height and size criteria before the sample data could be analyzed (Table 6 and Table 7). If these criteria were acceptable, the ratio of the blood to urine heterozygous alleles was calculated using the following formula: Ratio = ((urine 1 allele 1 peak height/urine 1 allele 2 peak height)/(blood 1 allele 1 peak height/blood 1 allele 2 peak height)). This ratio was then compared to cut-off values previously determined via the calculation of the ratios of 50 matched healthy blood and urine specimens.

### 4.12. Data Analysis

Data analysis was performed using GeneMapperID. “Positive” results indicated a variant or missing allele at the loci indicated. The target loci are known cancer markers. The loss of heterozygosity (LOH) at the loci indicated when one of the two alleles was missing from the urine sample at a heterozygous locus in the buccal sample. LOH can only be detected at heterozygous loci; hence, homozygous loci are uninformative. The overall MSA genotype was calculated as negative, LOH-low (1–2 markers), LOH-medium (3–4 markers), and LOH-high (>4 markers). Tumors typically acquire more LOH markers as they grow and advance in stage. Unless stated otherwise in the validation protocol, the following acceptance criteria were applied to determine the suitability of each assay.

### 4.13. ABI 3100 DNA Analyzer Acceptance Criteria

To be deemed negative (or with no LOH detected), a heterozygous locus had to have RFU peak heights between 200 RFUs and 100,000 RFUs with a peak height ratio between the cut-off values determined for each STR marker (see Table 7). To be deemed LOH/positive, the samples had to have at least one heterozygous locus with RFU peak heights that fell between 200 RFUs and 100,000 RFUs with a peak height ratio above or below the cut-off values determined for each STR marker. To be deemed non-informative, the sample had to be homozygous for the locus. A homozygous locus would have only one peak with sample RFU peak heights having to fall between 200 RFUs and 100,000 RFUs. If the sample did not meet the minimum or maximum RFU for the peak height or had no signal at all, it was deemed non-evaluable. A minimum of eight (8) loci had to be analyzed and had to meet the negative, LOH, or non-informative call in order to be deemed to have met the sample acceptance criteria. For failed samples, the analysis was repeated one time. If a sample failed in the second analysis, then it was deemed to have failed to meet the sample acceptance criteria and reported as quantity not sufficient (QNS).

## Data Availability

The datasets generated and/or analyzed during the current study are not publicly available, as there are no public repositories for this type of dataset. The data are available from the corresponding author upon reasonable request.

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
