# Peer review of "Results Obtained from a Pivotal Validation Trial of a Microsatellite Analysis (MSA) Assay for Bladder Cancer Detection through a Statistical Approach Using a Four-Stage Pipeline of Modern Machine Learning Techniques"

_ijms, 2023, doi:10.3390/ijms25010472_

Round 1

Reviewer 1 Report

Comments and Suggestions for Authors

This paper is a re-analysis of the data derived from a prospective study. The method is clearly described (although the novelty is low), the results are believable, and the discussion is comprehensive. I support the publication of this article after major revision. Some comments are as follows. 

1) Although LOH is a good potential biomarker, the MSI-associated LOH is not necessarily of diagnosis value and of high detection rate. 

2) From a genome-wide perspective, MSI variation is not prevalent in bladder cancer (BC), especially early BC. This is a well-known limit to this kind of biomarkers in BC detection. Therefore, the authors should provide more discussions on the advantage of their method in comparison with traditional panel-based method with respect to speed, accuracy and cost.

3) The quality of the figures should be improved. It seems that these figures were produced automatically with software. 

Comments on the Quality of English Language

The text should be polished to get it more readable.

Author Response

  1. Please see the rather extensive English editing with highlighted by yellow color in first and second  attachment, which we hope to satisfy the reviewer's comments.
  2. 1) Although LOH is a good potential biomarker, the MSI-associated LOH is not necessarily of diagnosis value and of high detection rate. 

    2) From a genome-wide perspective, MSI variation is not prevalent in bladder cancer (BC), especially early BC. This is a well-known limit to this kind of biomarkers in BC detection. Therefore, the authors should provide more discussions on the advantage of their method in comparison with traditional panel-based method with respect to speed, accuracy and cost. This is briefly mentioned and ways of improving MSI test has been discussed in page 11, Highlighted with green color in first attachment

    3) The quality of the figures should be improved. It seems that these figures were produced automatically with software. We apologize as this is our best effort figure based on our expertise

Reviewer 2 Report

Comments and Suggestions for Authors

Abstract:

The abstract successfully conveys the essence of the study and its findings. However, it is dense with jargon and could be simplified for broader accessibility. The claims of increased accuracy, sensitivity, and specificity need to be supported by a comparison to existing benchmarks in the field. There is also a lack of clarity around the statistical significance of the results and potential clinical implications.

Introduction:

The introduction adequately sets the context for the research, but it lacks a critical discussion of the literature, particularly in contrasting the proposed methods with existing techniques. The rationale behind selecting 16 MSA markers based on 9 publications could be seen as a potential selection bias, and this choice requires further justification.

Methods:

The methodological description is thorough, yet it could benefit from additional details on sample selection to ensure representativeness and mitigate selection bias. The use of a four-stage machine-learning pipeline is innovative but requires a clearer explanation of each stage to ensure reproducibility. Additionally, the decision to exclude certain samples from analysis could introduce bias, and the criteria for exclusion should be scrutinized.

Results:

The reported improvements in diagnostic metrics are promising. However, the manuscript would benefit from a more detailed discussion on the robustness of the results, including confidence intervals and measures of statistical significance. The potential for overfitting given the methodological approach, particularly the use of feature engineering and model-derived variable importance measures, is a concern that needs to be addressed.

Discussion:

The discussion offers a good synthesis of the findings, but it needs a stronger critical analysis of the limitations and potential biases, such as those arising from the analytical approach or the sample characteristics. There is also an opportunity to discuss the generalizability of the results to other populations and settings.

Overall Impression:

While the manuscript presents an interesting study with potential clinical applications, several issues need to be addressed. These include potential biases in marker selection, data analysis, and reporting of results. The methodological details need to be clarified to establish the study's validity and reliability. The language barrier issue is apparent and requires attention to ensure that the text is accessible to a wide scientific audience.

Recommendations:

Provide a clear rationale for the selection of MSA markers and discuss the potential for selection bias.

Elaborate on the sample selection criteria and discuss any exclusion criteria in detail to rule out sampling bias.

Clarify the methodology, particularly the machine learning stages, to ensure that the study can be replicated.

Discuss the statistical significance of the results, including confidence intervals and p-values.

Address the potential for overfitting and discuss strategies used to mitigate it.

Offer a more in-depth discussion on the limitations of the study and potential biases in the results.

Improve the language quality to meet publication standards.

Discuss the implications of the study's findings for clinical practice and future research directions.

Comments on the Quality of English Language

The English text demonstrates a need for improvement to reach publication standards. Technical terms should be clearly defined, and sentences should be structured to ensure clarity and coherence.

Author Response

Overall, there has been extensive language editing, which is marked with a yellow shadow. The title is shortened.

Abstract:

The abstract successfully conveys the essence of the study and its findings. However, it is dense with jargon and could be simplified for broader accessibility. The claims of increased accuracy, sensitivity, and specificity need to be supported by a comparison to existing benchmarks in the field. There is also a lack of clarity around the statistical significance of the results and potential clinical implications.

The last two sentences are in blue. The statistical significance and results are already appropriately reported in the abstract. Potential clinical implications are discussed in the discussion section as well which is the appropriate section.

Introduction:

The introduction adequately sets the context for the research, but it lacks a critical discussion of the literature, particularly in contrasting the proposed methods with existing techniques. The rationale behind selecting 16 MSA markers based on 9 publications could be seen as a potential selection bias, and this choice requires further justification.

On page 3, there is a comprehensive explanation of MSA as a viable technology and the importance of having any viable method for bladder cancer screening. At the end of our reference, we have provided the overall perspective of other commonly used screening tests (Ref 71). We have added specificity and sensitivity of cytology and cystoscopy. At the time of the EDRN study (2003-2008), the Johns Hopkins MSA markers were the only well-studied markers available.   The validity of these markers was later reproduced from another report, which was outlined in the discussion

Methods:

The methodological description is thorough, yet it could benefit from additional details on sample selection to ensure representativeness and mitigate selection bias. The use of a four-stage machine-learning pipeline is innovative but requires a clearer explanation of each stage to ensure reproducibility. Additionally, the decision to exclude certain samples from analysis could introduce bias, and the criteria for exclusion should be scrutinized.

The prospective study was designed to determine the efficacy of the panel of MSA markers for detecting bladder cancer and bladder cancer recurrence using the developed clinical MSA assay. The EDRN sample scheme and study objectives are added in page 14.

Results:

The reported improvements in diagnostic metrics are promising. However, the manuscript would benefit from a more detailed discussion on the robustness of the results, including confidence intervals and measures of statistical significance. The potential for overfitting given the methodological approach, particularly the use of feature engineering and model-derived variable importance measures, is a concern that needs to be addressed.

We have added references of methods and the used of ROC curves and AOC as the industry accepted measures. 

Discussion:

The discussion offers a good synthesis of the findings, but it needs a stronger critical analysis of the limitations and potential biases, such as those arising from the analytical approach or the sample characteristics. There is also an opportunity to discuss the generalizability of the results to other populations and settings.

On page 13, we have provided a clear rationale for selecting MSA markers and discuss the potential for selection bias. We have elaborated on the sample selection criteria and discussed any exclusion criteria in detail to rule out sampling bias. At this point, we can’t generalize our results to other cancers or populations.

Clarify the methodology, particularly the machine learning stages, to ensure that the study can be replicated.

We discussed the statistical significance of the results, including confidence intervals and p-values.

We address the potential for overfitting and discuss strategies used to mitigate it.

Discuss the implications of the study's findings for clinical practice and future research directions.

In page 13, para 465 to 472, we have summarized implications of the study's findings for clinical practice and future research directions.
